# Align before Adapt: Efficient and Generalizable Video Action Recognition with Text Corpus

## Abstract

Large-scale pre-trained visual-language models have achieved significant success in various video tasks. However, most existing methods follow an 'adapt then align' paradigm, where pre-trained image encoders are adapted to model video-level representations, which are then aligned to the semantics or one-hot labels of target actions. This paradigm overlooks the challenge of mapping from static images to complicated activity concepts. In this paper, we propose a novel and efficient 'align before adapt' paradigm. We introduce a token-merging strategy to the pre-trained image model, generating region-aware embeddings in a hierarchical manner. This enhances the visual-semantic alignment at a fine-grained level. Additionally, we align the region-aware embeddings with the text corpus of action-related entities, such as objects, body parts, primitive motions, and scenes. The embeddings of the aligned text entities serve as queries for the transformer-based video adapter, better aligning with the activity concepts in a video sequence. Our proposed framework achieves competitive performance and superior generalizability while significantly reducing computational costs. In fully-supervised scenarios, our method achieves 87.9% top-1 accuracy on Kinetics-400, using only 4947 GFLOPs. Furthermore, in 2-shot experiments, our method outperforms the previous state-of-the-art by 13.0% and 12.0% on HMDB-51 and UCF-101, respectively.

## 1 Introduction

Video action recognition is a fundamental task in the pursuit of intelligent video understanding. The recent trend of utilizing the visual-language pre-trained (VLP) models (Radford et al., 2021; Jia et al., 2021; Yu et al., 2022; Li et al., 2022a) have significantly advanced the research of action recognition (Wang et al., 2021; Ju et al., 2022; Pan et al., 2022; Ni et al., 2022; Lin et al., 2022; Yang et al., 2023). By lightly fine-tuning the model, VLP-based methods outperform the previous end-to-end network architectures, including two-stream networks (Simonyan & Zisserman, 2014; Wang et al., 2016; Zhou et al., 2018), 3D convolutional neural networks (Carreira & Zisserman, 2017; Feichtenhofer, 2020; Feichtenhofer et al., 2019; Hara et al., 2017; Qiu et al., 2017; Tran et al., 2015; 2018; Xie et al., 2018), and vision-transformer-based (ViT) networks (Bertasius et al., 2021; Fan et al., 2021; Liu et al., 2022; Patrick et al., 2021; Yan et al., 2022). Employing a pre-trained VLP model for action recognition can better encode the semantic meaning of items in images, even if they have very different visual appearances. This is very helpful in understanding human action and also explains why VLP models have achieved superior performance. As shown in Fig 1, the current VLP-based action recognition methods follow an "adapt then align" paradigm. They either introduces temporal interaction upon image representations or inserts temporal modules into pre-trained image encoders. However, the "adapt then align" paradigm potentially destruct the visual-semantic correspondences provided by VLP models, which will weaken the generalization ability of the action recognition for the following reasons: (1) Actions are complex concepts that involve multiple fine-grained atomic components, such as body parts, scenes, and objects. VLP models, with their inherent visual-semantic correspondences, enable capturing the semantic information of these components from various visual appearances. This facilitates accurate determination of action labels with improved interpretability (Fang et al., 2018). (2) Human-centric activities often share common components, implying that visual-semantic correspondences can be reused across different actions, even for those

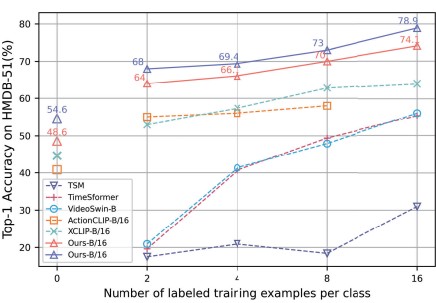

Figure 1: **Left:** Paradigm comparison between traditional adaptation approaches and our "Align before Adapt" method. **Right:** Zero-shot and few-shot performance comparison on HMDB-51 dataset. With the model trained on Kinetics-400, our method surpasses the previous state of the arts.

that were not included in the training set. This high level of reusability allows the model to quickly recognize new action categories.

In this paper, we propose an "align before adapt" paradigm. The paradigm first enhances the fine-grained visual-semantics alignment by adopting a token-merging strategy (Bolya et al., 2022), which hierarchically aggregates visual patches in a region-aware manner using a bipartite matching mechanism. Then, we align the region-aware visual features with a text corpus that incorporates action-related atomic entities, including objects, body parts, primitive motions, and scenes. The corpus is generated by an automatic entity extraction procedure and can be reused for different action recognition datasets. Each visual feature is matched with the text embeddings of entities in the corpus using vector similarity, and the entity with the highest similarity is regarded as aligned with the visual embedding. We leverage the aligned entities to guide the video representation adaption. Specifically, the text embeddings of the aligned entities serve as queries for the transformer-based video adapter, improving the alignment between the frame-level visual embedding and video-level activity concepts.

In the "align before adapt" paradigm, both region-aware visual-semantic alignment and aligned-entity-guided video adaptation benefit the video action recognition. The embeddings of the token-merging patches form "one-to-one" relationships between local visual patterns and the text corpus, resembling the measures of visual grounding (Ghiasi et al., 2022; Li et al., 2022c). The text embeddings of the aligned entities serve as regularization for adaptation, preserving the visual-semantic correspondences from VLP. The overall paradigm with the token merging strategy helps us achieve better generalization ability with low computational complexity. It improves our framework by 6.3% top-1 accuracy on the HMDB-51 dataset under the 2-shot configuration but requires 23% less computational cost with ViT-base backbone. In summary, our contributions are threefold:

- We propose an "align before adapt" paradigm that leverages the region-aware visual-semantic alignment and the aligned-entity guided video adaptation. The paradigm preserves the visual-semantic alignment of VLP during the video representation adaption, achieving better interpretability and generalization ability.

- We apply a token merging strategy on pre-trained image models. The embeddings generated from merged patches form "one-to-one" relationships between local visual patterns and the text corpus. It can also reduce the computational complexity of the transformer-based image encoder.

- Extensive experiments under various learning configurations are conducted. Besides demonstrating competitive performance with low computational complexity (surpasses the current leading approach with the same VLP backbone by *0.4%* top-1 accuracy while requiring *55%* fewer GFLOPs), our method reveals superior generalizability due to the reusable text entities (surpasses the previous state-of-the-art by more than *10%*).

## 2 RELATED WORK

**Large Scale Visual-Language pretraining.** In the past few years, the surge of large-scale visual-language pre-trained (VLP) models (Su et al., 2020; Li et al., 2020a; Radford et al., 2021; Jia et al., 2021; Li et al., 2022a; Zeng et al., 2022) have revolutionized multiple fields of computer vision,

including image classification, captioning (Yu et al., 2022), grounding (Li et al., 2022c), image-text retrieval, and so on. With the availability of massive amounts of web-scale visual-text paired data, these models learn cross-modal representations through masked language modeling and/or contrastive learning. Specifically, one of the most representative works, CLIP (Radford et al., 2021) , is trained on 400M data following a contrastive manner, and shows remarkable performance on zero-shot image classification. The success of VLP models inspires the "fine-tuning" trend on multiple downstream tasks, such as open-vocabulary detection (Gu et al., 2022), segmentation (Wang et al., 2022; Xu et al., 2023; Ghiasi et al., 2022), caption (Mokady et al., 2021), summarization (Narasimhan et al., 2021), generation (Ramesh et al., 2022), etc. Our method adopts CLIP as the backbone for video action recognition tasks under fully-supervised, few-shot, and zero-shot scenarios.

**Video Action Recognition.** The prosperity of deep learning has sparked various works for effective video action recognition. Initially, there were two directions of methods: two-stream 2D CNNs (Zhou et al., 2018; Wang et al., 2016; Simonyan & Zisserman, 2014) that process and spatial and temporal context parallelly, and 3D CNNs (Tran et al., 2015; 2018; Qiu et al., 2017; Xie et al., 2018; Feichtenhofer et al., 2019; Feichtenhofer, 2020) that factorize the convolution across spatial and temporal dimensions simultaneously. Later transformer-based approaches, including ViViT (Arnab et al., 2021), Timesformer (Bertasius et al., 2021), and VideoSwin (Liu et al., 2022) outperformed the convolutional methods, by better capturing long-term dependencies through scalable self-attention mechanisms. More Recently, leveraging available VLP models such as CLIP (Radford et al., 2021) and Florence (Yuan et al., 2021), has become a data-friendly trend. EVL (Lin et al., 2022), ST-Adapter (Pan et al., 2022), and AIM (Yang et al., 2023) add lightweight modules to the fixed CLIP backbone for close-set recognition tasks, while ActionCLIP (Wang et al., 2021) and X-CLIP (Ni et al., 2022) propose frameworks that enable adaptation to new scenarios. While all of the above methods focus on adapting the visual branch of VLP models to the video directly, our approach introduces early, grounded visual-semantic alignments before the adaptation step. This effectively bridges the gap of mapping with complicated activity semantics during video representation learning.

**Region-Aware Perception for Vision Transformer.** In recent research on ViT architectures, it has been well-studied that capturing fine-grained patterns in visual signals improves representation learning. Various approaches, such as Swin Transformer (Liu et al., 2022), Region ViT (Chen et al., 2022a), and GCViT (Hatamizadeh et al., 2023), propose incorporating multi-scale attention into the ViT to achieve better performance in various downstream tasks including recognition, detection, and segmentation. With the rise of visual-language pretraining, GLIP (Li et al., 2022c) suggests learning better instance-level language-aware representations through grounded image-text data, while FILIP (Yao et al., 2022) and Dense CLIP (Rao et al., 2022) focus on introducing patch-level contrastive losses. These works have shown impressive progress in open-world scenarios. In contrast to methods that require additional structures, data, or supervision, we adopt ToMe in our image encoder. ToMe is computationally efficient and does not require any additional parameters. It incorporates a soft bipartite matching strategy and a proportional attention mechanism to hierarchically merge visual tokens in a region-aware manner. We utilize the merged tokens to achieve fine-grained visual-semantic alignments for video action recognition.

## 3 METHODOLOGY

Our method aims to learn discriminative and transferable video representations for action recognition. An overview of our proposed method is illustrated in Fig 2. We begin by constructing text corpus based on action label sets offline (Sec. 3.1). Our "align before adapt" framework is introduced in the following two sections: exploring fine-grained visual-semantic alignments in Sec. 3.2; And leveraging aligned semantic embeddings for the subsequent adaptation of the video representation learning (Sec. 3.3) The training details are introduced in Sec 3.4.

### 3.1 ACTION-RELATED TEXT CORPUS CONSTRUCTION

Drawing inspiration from cognitive science and recent research (Kurby & Zacks, 2008; Zacks et al., 2001; Fang et al., 2018; Li et al., 2020b), we believe that perceiving and leveraging intermediate spatiotemporal-varied patterns, such as bodies, objects, and scenes, can greatly mitigate the difficulty of understanding activity concepts. Thanks to the VLP models, these patterns can be linguistically

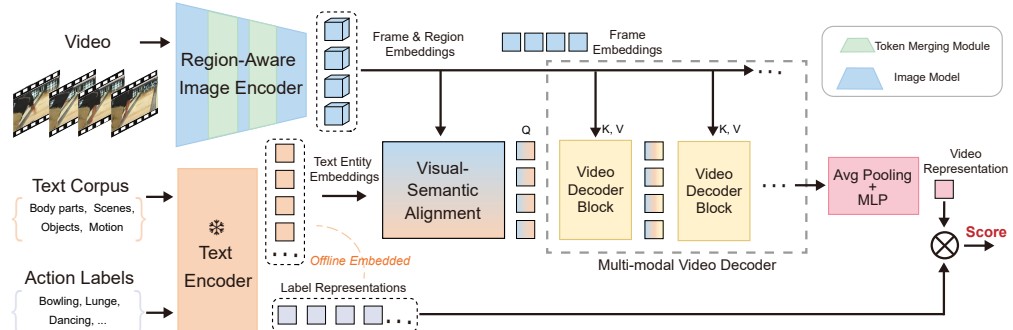

Figure 2: An overview of our framework: we input a video clip and offline embedded text corpus to generate a video representation (in pink). This video representation is supervised with the goal of maximizing the similarity score with the text representation of the correct action label.

expressed and perceived based on their similarities with visual representations in the embedding space. We construct a knowledge base for these patterns, referred to as "text corpus".

Empowered by large-scale language models, we design an automated process to construct the action-related text corpus. To generate a reusable text corpus, we first collect a set of action labels from several recognition datasets such as Carreira et al. (2018); Kuehne et al. (2011); Soomro et al. (2012). The action-related text corpus is then generated according to the following steps: We employ NLTK (Bird et al., 2009) to remove stop words and split single words or phrases from the descriptions as basic units. As a supplement, we utilize the API of ChatGPT (OpenAI, 2022) with handcraft prompts to uncover implicit units behind action names. (2) Then we utilize WordNet (Miller, 1995) and prompt ChatGPT to automatically generate a series of descriptions as candidates for each unit. (3) To filter out the appropriate descriptions, we employ word sense disambiguation techniques, including the Lesk algorithm (Basile et al., 2014) and the T5 language model (Wahle et al., 2021). These filtered descriptions are paired with correlated basic units and added to the text corpus in the form of `text entity={unit, description}`. In addition, the body parts, such as `head`, `hands`, and `feet` are added to the corpus by default. They are involved in most human activities. The details and an illustration of the process are provided in Appendix A.

The produced text corpus is denoted by $\mathcal{S} = \{s_i\}_{i=1}^{K}$, where $K$ is the number of the **text entities**. We employ the text encoder of CLIP to encode every text entity in the text corpus to a $d$-dimension embedding offline. The embeddings of all the text entities can be formulated as $\mathbf{S} \in \mathbb{R}^{K \times d}$.

## 3.2 VISUAL-SEMANTIC ALIGNMENT

**Region-aware image encoding.** To fully explore the fine-grained visual-semantic alignments, we first aim to adaptively perceive and encode specific instances in video frames. In Fig 3a, we introduce a region-aware image encoder, which consists of two parts: a ViT initialized by CLIP is used to encode the frames; Token merging modules (Bolya et al., 2022) are incorporated into each transformer encoder block of ViT to merge intermediate patch-wise tokens hierarchically.

Given a frame $\mathbf{I} \in \mathbb{R}^{H \times W \times C}$, the ViT splits the frame into $N$ non-overlapping patches and projects them into $d$-dimension embeddings, forming a sequence $\{\mathbf{e}_i\}_{i=1}^{N}$. The sequence is prepended with a learnable $[class]$ embedding $\mathbf{e}_{cls}$ and added with position embeddings $\mathbf{E}_{pos}$, resulting in $\mathbf{E}_0$:

$$\mathbf{E}_0 = [\mathbf{e}_{cls}^\top, \mathbf{e}_1^\top, \mathbf{e}_2^\top, ..., \mathbf{e}_N^\top] + \mathbf{E}_{pos} \quad \in \mathbb{R}^{(N+1) \times d}. \tag{1}$$

$\mathbf{E}_0$ serves as the input for the sequence of encoder blocks. Each encoder block is composed of a multi-head self-attention (MSA) and an MLP layer, along with a token merging (ToMe) module and Layernorms (LN). Particularly, during the token merging process, a soft bipartite matching algorithm (Bolya et al., 2022) is employed to find the most similar $r$ pairs of embeddings based on cosine similarities. The embeddings of each pair are merged into one, resulting in the total reduction of $r$. The *weights* $w$ of embeddings (how many embeddings have the current embedding incorporated)

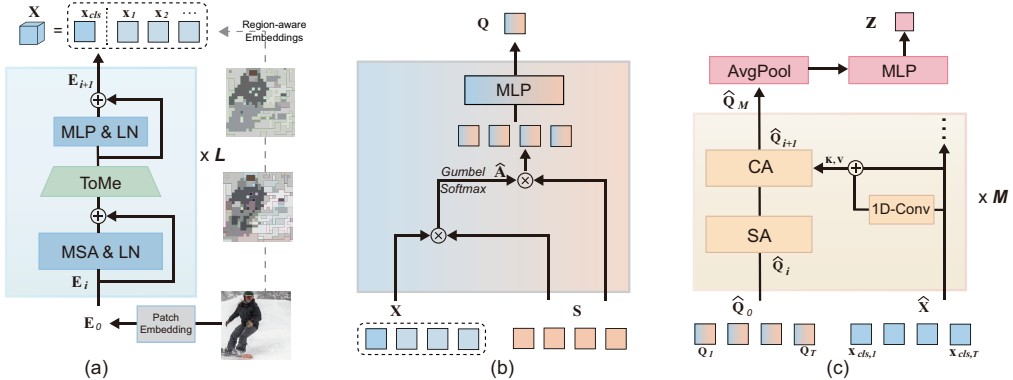

Figure 3: Detailed network components: (a) The region-aware image encoder includes a ViT with plug-in token merging modules. (b) The semantic alignment module obtains the aligned query in a softmax-weight-sum manner. and (c) shows the multi-modal video decoder, with its decoding block containing a stack of hybrid modules composed of attention layers and 1D temporal convolution.

are tracked and utilized for merging operation and MSA. The procedure can be formulated as:

$$\mathbf{E}_i' = \mathbf{E}_{i-1} + \mathrm{MSA}(\mathrm{LN}(\mathbf{E}_{i-1}), w_i), \quad \mathbf{E}_i'' = \mathrm{ToMe}(\mathbf{E}_i', w_i),$$
$$\mathbf{E}_i = \mathbf{E}_i'' + \mathrm{MLP}(\mathrm{LN}(\mathbf{E}_i'')), \quad \mathbf{E}_i \in \mathbb{R}^{(N+1-i\times r)\times d}, \tag{2}$$

where $i$ denotes the index of transformer layers, and $w_i$ is the embedding weights in the current block. Notably, the MSA layer adds an extra $\log w_i$ term in the softmax calculation. The ToMe module merges the tokes (excludes the $[class]$ embedding) by averaging them with the weights $w_i$. We denote the final output of the L-block image encoder as $\mathbf{X}$, which contains the frame-level [class] embeddings and region-aware (remaining merged ones) embeddings:

$$\mathbf{X} = \mathbf{E}_L = [\mathbf{x}_{cls}^\top, \mathbf{x}_1^\top, \mathbf{x}_2^\top, ...] \in \mathbb{R}^{(N+1-L\times r)\times d}. \tag{3}$$

The right side of Fig. 3a visualizes the merging procedure along the transformer blocks. The visual patches with the same color and border are merged into one and form region-aware embeddings.

**Fine-grained visual-semantic alignments.** We explore frame-level and grounded visual-semantic correlations based on the text corpus and obtained visual embeddings. Specifically, the text entities from the corpus are adaptively aligned with each input frame according to the similarities in the embedding space. As depicted in Fig 3b, we start by calculating the similarity matrix $\mathbf{A}$ between the embeddings of text entities $\mathbf{S}$ and the visual embeddings of the frame $\mathbf{X}$. This is achieved by applying Gumbel-Softmax (Jang et al., 2017; Maddison et al., 2017) operation over $\mathbf{S}$:

$$w_{i,j} = \frac{\langle \mathbf{x}_i, \mathbf{s}_j \rangle}{\| \mathbf{x}_i \| \cdot \| \mathbf{s}_j \|}, \quad \mathbf{A}_{i,j} = \frac{\exp(w_{i,j}/\tau + \gamma_j)}{\sum_{k=1}^{K} \exp(w_{i,k}/\tau + \gamma_k)}, \tag{4}$$

where $\mathbf{x}_i \in \mathbb{R}^d$ and $\mathbf{s}_j \in \mathbb{R}^d$ are $i$th and $j$th embeddings of $\mathbf{X}$ and $\mathbf{S}$, respectively. $w_{i,j}$ is cosine similarity between $\mathbf{x}_i$ and $\mathbf{s}_j$. $A_{i,j}$ is gumbel-softmax calculation of $w_{i,j}$ over $K$-class $\mathbf{S}$ with the temperature term $\tau$ and i.i.d random samples $\gamma_i$ drawn from the $Gumbel(0,1)$ distribution. We then align each visual embedding with a text entity in a *hard-assignment* (one-to-one) manner, which is targeted at alleviating ambiguity of semantics and achieved by taking one-hot operations of $\mathbf{A}_{argmax}$ over the all the text entities. Since the one-hot operation is not differentiable, we utilize the trick of van den Oord et al. (2017); Xu et al. (2022):

$$\widehat{\mathbf{A}} = one\text{-}hot(\mathbf{A}_{argmax}) + \mathbf{A} - sg(\mathbf{A}), \tag{5}$$

where $sg$ is the stop gradient operator. $\widehat{\mathbf{A}} \in \mathbb{R}^{(N+1-L\times r)\times K}$ assign the most correlated text entity for each frame-level or region-aware embedding with dominating weights while keeping differentiable. It is noteworthy that, to validate the precision and interpretability of visual-semantic alignments, in Fig. 4 left, we visualize the correspondence between region-aware embeddings and text entities. The aligned semantic embedding $\mathbf{Q}$ for frame $\mathbf{I}$ is calculated by weighted summing $\widehat{\mathbf{A}}$ and $\mathbf{S}$, followed by an MLP layer to reduce the dimensions from $\mathbb{R}^{(N+1-L\times r)\times d}$ to $\mathbb{R}^{1\times d}$:

$$\mathbf{Q} = \mathrm{MLP}(\mathbf{A}\mathbf{S}). \tag{6}$$

### 3.3 Video Representation Adaptation

Providing factorized text knowledge behind actions, the aligned semantic embeddings are leveraged as "queries" for the adaptation of learning video representation. Given a video with $T$ frames $[\mathbf{I}_1, \mathbf{I}_2, ..., \mathbf{I}_T]$, we can obtain the frame-level embeddings $\widehat{\mathbf{X}} \in \mathbb{R}^{T \times d}$ and corresponding aligned semantic embeddings $\widehat{\mathbf{Q}} \in \mathbb{R}^{T \times d}$ according to Eq. 3 and Eq. 6, respectively:

$$\widehat{\mathbf{X}} = [\mathbf{x}_{cls,1}^\top, ..., \mathbf{x}_{cls,i}^\top, ..., \mathbf{x}_{cls,T}^\top], \quad \widehat{\mathbf{Q}} = [\mathbf{Q}_1, ..., \mathbf{Q}_i, ..., \mathbf{Q}_T], \tag{7}$$

where $\mathbf{x}_{cls,i}$ represents the frame-level embedding of the $i$th frame. We propose a multi-modal video decoder, as shown in Fig 3c. This decoder includes a sequence decoding block that consists of a 1D-convolution module, a self-attention (SA) module, and a cross-attention (CA) module. The attention modules function in the same way as the ones in the transformer (Vaswani et al., 2017). The SA module and the 1D-convolution module serve for temporal interactions among the aligned semantic embeddings $\widehat{\mathbf{Q}}$ and visual embeddings $\widehat{\mathbf{X}}$, respectively. The CA module utilizes semantic embeddings as queries to attend to the visual embeddings across the frames, thus facilitating communication between the two modalities. The procedure of the video decoder can be formulated as:

$$\widehat{\mathbf{X}}_i' = \widehat{\mathbf{X}} + \text{1D-Conv}^i(\widehat{\mathbf{X}}), \quad \widehat{\mathbf{Q}}_i' = \text{SA}^i(\widehat{\mathbf{Q}}_{i-1}), \quad \widehat{\mathbf{Q}}_i = \text{CA}^i(\widehat{\mathbf{Q}}_i', \widehat{\mathbf{X}}_i'), \quad i = 1, ..., M, \tag{8}$$

where $i$ indicates the block index of the video decoder. The initialized query $\widehat{\mathbf{Q}}_0 = \widehat{\mathbf{Q}}$, and the output query after $M$ blocks is $\widehat{\mathbf{Q}}_M$. We obtain the final video representation $\mathbf{z}$ by applying Average Pooling and MLP layer over the $\widehat{\mathbf{Q}}_M$ sequentially along the temporal dimension and the feature channel:

$$\mathbf{z} = \text{MLP}(\text{AvgPool}(\widehat{\mathbf{Q}}_M)). \tag{9}$$

### 3.4 Training Details

**Loss function.** Our proposed network aims to maximize the similarity between video representations and textual representations of corresponding action labels. Specifically, we utilize the frozen text encoder of CLIP to perform prompt ensembling for action labels with a bunch of handcraft templates (Ni et al., 2022). Given the representations of the $i$th action $\mathbf{c}_i$ and $n$th video $\mathbf{z}_n$ (as described in Eq. 9), the loss function can be implemented by the cross-entropy loss:

$$\mathcal{L} = -\frac{1}{N} \sum_{n=1}^{N} \sum_{i=1}^{I} y^{i,n} \log \left( \frac{\exp(\mathbf{c}_i^\top \mathbf{z}_n)}{\sum_{j=1}^{I} \exp(\mathbf{c}_j^\top \mathbf{z}_n)} \right). \tag{10}$$

The training set has $N$ videos belonging to the $I$ actions. If the $n$th video belongs to the $i$th action, $y^{i,n}$ equals 1; otherwise, $y^{i,n}$ equals 0.

**Network training.** The ViT backbone in the region-aware image encoder is initialized by CLIP, while the token merging module is parameter-free with the reduction number $r$ to be 8. The number of blocks in the multi-modal video decoder is set to 4 and 6 for ViT-B and ViT-L backbones, respectively. We adopt an AdamW optimizer for network parameter training with initial learning rates of $8 \times 10^{-6}$ for the ViT backbone and $8 \times 10^{-5}$ for the remaining parts. The networks are trained with 30 epochs (including a five-epoch warmup) and a weight decay of 0.001 w.r.t. a cosine schedule. The input video follows the main sparse sampling method (Wang et al., 2016) and augmentation strategy (Ni et al., 2022) with a frame resolution $224 \times 224$. Experiments are conducted with 8 32GB-V100-GPUs.

## 4 Experiments

**Datasets.** Our proposed method is evaluated on four widely used video action recognition datasets: Kinetics-400 (Kay et al., 2017), Kinetics-600 (Carreira et al., 2018), UCF-101 (Soomro et al., 2012), and HMDB-51 (Kuehne et al., 2011). Kinetics-400 consists of approximately 240k training and 20k validation videos, covering 400 classes, with each clip spanning around 10 seconds. Kinetics-600 is an extension of Kinetics-400, including around 410k training and 29k validation videos for 600 classes. UCF-101 contains 13,320 video clips with 101 classes, and HMDB-51 consists of 7,000 videos with 51 classes. We conduct fully-supervised experiments on Kinetics-400 and Kinetics-600, and for Kinetics-600, UCF-101, and HMDB-51, we perform few-shot and zero-shot experiments using a pre-trained model from Kinetics-400.

Table 1: Comparison to state-of-the-art on Kinetics-400. Views are denoted with *"input frames ×
spatial crops × temporal clips."* For image-language approaches, parameters in the text branch are
not counted. * indicates pretraining with a video-text collection.

| Method | Pretrain | Top-1 | Top-5 | GFLOPs | Views | #Param.(M) |
|---|---|---|---|---|---|---|
| *Methods with ImageNet or web-scale image pretraining* | | | | | | |
| MViTv1-B (Fan et al., 2021) | - | 81.2 | 95.1 | 4095 | 64×3×3 | 36.6 |
| Uniformer-B (Li et al., 2022b) | IN-1k | 83.0 | 95.4 | 3108 | 32×4×3 | 50.0 |
| TimeSformer-L (Bertasius et al., 2021) | IN-21k | 80.7 | 94.7 | 7140 | 64×1×3 | 121.4 |
| VideoSwin-L (Liu et al., 2022) | IN-21k | 83.1 | 95.9 | 7248 | 32×4×3 | 200.0 |
| ViViT-H/16 (Arnab et al., 2021) | JFT-300M | 84.9 | 95.8 | 48916 | 32×4×3 | 647.5 |
| *Methods with web-scale language-image pretraining* | | | | | | |
| MTV-H/16 (Yan et al., 2022) | WTS-17B* | 89.1 | 98.2 | 45537 | 32×4×3 | - |
| PromptingCLIP-B/16 (Ju et al., 2022) | CLIP-400M | 76.9 | 93.5 | - | 16×5×1 | 95.5 |
| ActionCLIP-B/16 (Wang et al., 2021) | CLIP-400M | 83.8 | 97.1 | 16890 | 32×10×3 | 105.2 |
| ST-Adapter-L/14 (Pan et al., 2022) | CLIP-400M | 87.2 | 97.6 | 8248 | 32×3×1 | - |
| EVL-L/14 (Lin et al., 2022) | CLIP-400M | 87.3 | 97.6 | 8088 | 32×3×1 | 357.9 |
| AIM-L/14 (Yang et al., 2023) | CLIP-400M | 87.5 | 97.7 | 11208 | 32×3×1 | 341 |
| X-CLIP-L/14 (Ni et al., 2022) | CLIP-400M | 87.1 | 97.6 | 7896 | 8×4×3 | 451.2 |
| *Our method* | | | | | | |
| ALT-B/16 | CLIP-400M | 83.6 | 96.2 | 330 | 8×1×3 | 124.6 |
| ALT-B/16 | CLIP-400M | 84.6 | 96.7 | 657 | 16×1×3 | 124.6 |
| ALT-B/16 | CLIP-400M | 85.4 | 96.7 | 1308 | 32×1×3 | 124.6 |
| ALT-L/14 | CLIP-400M | 86.7 | 97.4 | 1245 | 8×1×3 | 427.1 |
| ALT-L/14 | CLIP-400M | 87.6 | 97.6 | 2478 | 16×1×3 | 427.1 |
| ALT-L/14 | CLIP-400M | 87.9 | 97.7 | 4947 | 32×1×3 | 427.1 |

Figure 4: **Left:** Visualization of visual-semantic correspondences with the tool (Chen et al., 2022b). For each
row: Column (2) visualizes the visual correspondence to text entities generated by the action label; Column
(3) visualizes region-aware embeddings under ToMe; Column (4) and (5) show the two of the fine-grained
corresponding visual patterns to specific text entities, which are geometrically consistent with Column (3).
**Right:** Visualization of Accuracy v.s. FLOPs performance.

## 4.1 FULLY SUPERVISED COMPARISON

**Settings.** We conduct fully-supervised experiments on Kinetics-400. Each video clip is sampled with
8, 16, or 32 frames. Two variants of the network, namely ALT-B/16 and ALT-L/14, employ ViT-B/16
and ViT-L/14, respectively. The results on Kinetics-600 are exhibited in the supplementary materials.

**Results.** In Tab. 1, we compare with the state-the-of-art methods on Kinetics-400 with the input
resolution 224×224. Taking eight sampled frames of each video as input, our method (ALT-B/16)
achieves *83.6%* top-1 accuracy with only 330 GFLOPs. When the input frames increase to 32,
ALT-B/16 surpasses the performance of ViViT-H/16 (Arnab et al., 2021), which takes more than
30× computation cost (1308 vs. 48916 GFLOPs). By employing the larger backbone, ALT-L/14
achieves superior performance with *87.9%* top-1 accuracy among CLIP-400M pretraining works
and significant computational advantage ( *0.4%* higher than AIM (Yang et al., 2023) but *55%* fewer

Table 2: Few-shot comparisson on HMDB-51 and UCF-101.

| Method | Frames | HMDB-51 | | | | UCF-101 | | | |
|---|---|---|---|---|---|---|---|---|---|
| | | $K$=2 | $K$=4 | $K$=8 | $K$=16 | $K$=2 | $K$=4 | $K$=8 | $K$=16 |
| TSM (Lin et al., 2019) | 32 | 17.5 | 20.9 | 18.4 | 31.0 | 25.3 | 47.0 | 64.4 | 61.0 |
| TimeSformer (Bertasius et al., 2021) | 32 | 19.6 | 40.6 | 49.4 | 55.4 | 48.5 | 75.6 | 83.7 | 89.4 |
| VideoSwin-B (Liu et al., 2022) | 32 | 20.9 | 41.3 | 47.9 | 56.1 | 53.3 | 74.1 | 85.8 | 88.7 |
| ActionCLIP (Wang et al., 2021) | 8 | 55.0 | 56.0 | 58.0 | - | 80.0 | 85.0 | 89.0 | - |
| X-CLIP-B/16 (Ni et al., 2022) | 32 | 53.0 | 57.3 | 62.8 | 64.0 | 76.4 | 83.4 | 88.3 | 91.4 |
| X-Florence (Ni et al., 2022) | 32 | 51.6 | 57.8 | 64.1 | 64.2 | 84.0 | 88.5 | 92.5 | 94.8 |
| ALT-B/16 | 32 | 64.0 | 66.1 | 70.0 | 74.1 | 93.2 | 95.3 | 96.3 | 97.1 |
| ALT-L/14 | 32 | **68.0** | **69.4** | **73.0** | **78.9** | **96.0** | **97.3** | **98.0** | **98.1** |

Table 3: Zero-shot on HMDB-51 & UCF-101.

| Method | HMDB-51 | UCF-101 |
|---|---|---|
| MTE (Xu et al., 2016) | 19.7±1.6 | 15.8±1.3 |
| ASR (Wang & Chen, 2017) | 21.8±0.9 | 24.4±1.0 |
| ZSECOC (Qin et al., 2017) | 22.6±1.2 | 15.1±1.7 |
| UR (Zhu et al., 2018) | 24.4±1.6 | 17.5±1.6 |
| TS-GCN (Gao et al., 2019) | 23.2±3.0 | 34.2±3.1 |
| ER-ZSAR (Chen & Huang, 2021) | 35.3±4.6 | 51.8±2.9 |
| ActionCLIP (Wang et al., 2021) | 40.8±5.4 | 58.3±3.4 |
| X-CLIP-B/16 (Ni et al., 2022) | 44.6±5.2 | 72.0±2.3 |
| ALT-B/16 | 48.6±1.2 | 76.7±0.9 |
| ALT-L/14 | **54.6±1.1** | **83.0±1.1** |

Table 4: Zero-shot on Kinetics-600.

| Method | Top-1 | Top-5 |
|---|---|---|
| DEVISE (Frome et al., 2013) | 23.8±0.3 | 51.0±0.6 |
| ALE (Akata et al., 2016) | 23.4±0.8 | 50.3±1.4 |
| SJE (Akata et al., 2015) | 22.3±0.6 | 48.2±0.4 |
| ESZSL (Romera-Paredes & Torr, 2015) | 22.9±1.2 | 48.3±0.8 |
| DEM (Zhang et al., 2017) | 23.6±0.7 | 49.5±0.4 |
| GCN (Ghosh et al., 2020) | 22.3±0.6 | 49.7±0.6 |
| ER-ZSAR (Chen & Huang, 2021) | 42.1±1.4 | 73.1±0.3 |
| X-CLIP-B/16 (Ni et al., 2022) | 65.2±0.4 | 86.1±0.8 |
| ALT-B/16 | 68.1±0.6 | 88.1±0.4 |
| ALT-L/14 | **72.7±0.4** | **90.7±0.3** |

GFLOPs). It is noteworthy that the leading method MTV-H (Yan et al., 2022) adopts larger-scale pretraining data (70M video-text pairs with about 17B images) and consumes $9\times$ GFLOPs. As shown in Fig 4 right, we visualize the performance of some representative works. Our approach achieves higher accuracy while using fewer inference GFLOPs, setting new Pareto frontiers.

## 4.2 FEW-SHOT COMPARISONS

**Settings.** We evaluate our few-shot experiments on the HMDB-51 and UCF-101 datasets. To construct the training set, we randomly sample 2, 4, 8, and 16 videos from each class, and we set the frame number in each video to either 8 or 32. Following the protocols of X-CLIP (Ni et al., 2022), we use the first split of the test set for evaluation and report the results based on single-view inference.

**Results.** Tab. 2 shows the performance comparison on $K$-shot learning. Our method significantly outperforms methods based on image pretraining. For instance, when $K$=2, ALT-B/16 surpasses VideoSwin-B (Liu et al., 2022) by *43.1%* on HMDB-51 and *39.9%* on UCF-101. Among the image-language pertaining models. When $K$=4, the ALT-L/14 surpasses the previous state of arts by *11.6%* and *8.8%* on HMDB-51 and UCF-101, respectively. The lead is consistent along to $K$=16 and continues to expand when switching to ALT-L/14, highlighting the effectiveness of our paradigm.

## 4.3 ZERO-SHOT COMPARISONS

**Settings.** For zero-shot performance evaluation, we pretrain our ALTs on Kinetics-400 with 32 frames and utilized the same protocol as (Ni et al., 2022): For HMDB-51 and UCF-101, we conducted experiments using the three provided splits. Regarding Kinetics-600, the test set is constructed by randomly selecting 160 categories from the 220 categories that are distinct from those in Kinetics-400 three times. We report single-view results in the format of *"average accuracy ± standard deviations."*

**Results.** We present the zero-shot results in Tab. 3 and Tab. 4. ALT-B/16 outperforms X-CLIP-B/16 by *4.0%*, *4.7%*, and *2.9%* in terms of top-1 accuracy on HMDB-51, UCF-101, and Kinetics-600, respectively. We attribute the superiority to the utilization of the text corpus, whose factorized and reusable semantics mitigate the difficulty of adapting our model to a new scenario.

## 4.4 ABLATION STUDY

We employ ALT-B/16 to conduct detailed ablation experiments. By default, with 8 frames per sample, the fully-supervised experiments are evaluated on Kinetics-400. Taking 32 frames as input, the few-shot experiments are conducted on the first split of HMDB-51, and the zero-shot evaluation is on the first split of the validation set of UCF-101. Results are obtained under single-view inference.

Table 5: Effect of proposed components. *Align*: Semantic alignment. *ToMe*: Token Merging. *Region*: Utilize merged region-aware visual tokens for *Align*.

| Align. | ToMe. | Region. | Fully. | 2-shot | 0-shot |
|--------|-------|---------|--------|--------|--------|
| -      | -     | -       | 81.7   | 53.0   | 72.0   |
| ✓      | -     | -       | **82.6** | 57.7 | 75.0   |
| ✓      | ✓     | -       | 82.0   | 54.1   | 73.3   |
| ✓      | ✓     | ✓       | 82.4   | **64.0** | **76.7** |

Table 6: Linear evaluation, Top-1 Acc.(%) reported in a single view of 32 frames, with ALT-B pre-trained on Kinetics-400. Splits are provided by datasets.

| Dataset | Split 1 | Split 2 | Split 3 | Average |
|---------|---------|---------|---------|---------|
| UCF-101 | 95.6    | 95.8    | 96.1    | 95.83   |
| HMDB-51 | 73.8    | 73.5    | 74.0    | 73.77   |

Table 7: Effect of different training strategies. *S*: spatial, *M*: MLP, *T*: temporal, *Param.*: # parameters.

| Method | Top-1 Acc.(%) | GFLOPs | Param.(M) | Tunable Param.(M) |
|--------|---------------|--------|-----------|-------------------|
| Frozen | 81.1 | 110 | 125 | **38** |
| S-adapter | 81.4 | 116 | 129 | 42 |
| SM-adapter | 81.5 | 123 | 132 | 45 |
| STM-adapter | 82.0 | 163 | 136 | 49 |
| Fine-tune | **82.4** | **110** | **125** | 125 |

Table 8: Trade-off between single-view efficiency & accuracy. $r$: the number of tokens to reduce in each transformer block

| $r$ | GFLOPs | Top-1 Acc.(%) |
|-----|--------|---------------|
| 0   | 141    | 82.6          |
| 4   | 129    | 82.4          |
| 8   | 110    | 82.4          |
| 13  | 86     | 81.9          |

**Component analysis.** To investigate the effectiveness of the proposed components, (1) we treat the X-CLIP-B/16 (w/o text prompts) as the baseline, which introduces a cross-frame module inside the CLIP image encoder and achieves *81.7%* top-1 accuracy on Kinetics-400. Results are presented in the first row of Tab. 5. (2) By replacing the cross-frame module with our video decoder and introducing semantic alignments, we improve the baseline to *82.6%*, demonstrating the effectiveness of our "align before adapt" paradigm. (3) We leverage ToMe (Bolya et al., 2022) in the image encoder, resulting in fewer computations but nonnegligible accuracy drops. (4) By further utilizing the merged region-aware visual embeddings for fine-grained semantic alignments, the losses from token merging are alleviated in fully-supervised experiments and even over-compensated in few/zero-shot scenarios.

**Linear evaluation on learned representations.** To investigate the quality of our learned video representations, we conduct linear probe experiments on UCF101 and HMDB-51 with a frozen ALT-B/16 pre-trained on Kinetics-400. The video representations extracted by the ALT-B/16 are fixed and fed into a tunable linear classification layer. Under the supervision of one-hot labels, we adopt the same AdamW schedule with Sec 3.4) and a learning rate $4 \times 10^4$. The top-1 accuracies are reported in Tab. 6, demonstrating the discrimination and generalization capabilities of our framework.

**Training strategy of image encoder.** Our method requires fine-tuning the pre-trained image encoder. By contrast, as shown in Tab. 7, (1) when locking the image encoder during training, fewer parameters demand tuning while the accuracy decreases to *81.1%*. (2)-(4) Based on the "frozen" setting, we leverage approaches provided by AIM (Yang et al., 2023), which introduces adapters into the image encoder. The performances are improved by stacking spatial, temporal, and MLP adapters in the transformer blocks. It is noteworthy that the STM method computes the attention layers twice, therefore significantly increasing the computational complexity.

**Efficiency and accuracy trade-off.** By default, we set the number of token reductions per block in the image encoder $r$ to 8. Here we further investigate the performance of varying $r$. As shown in Tab. 8, our method achieves the highest accuracy in fully-supervised experiments without the token merging strategy (also no region-aware semantic alignment.) As $r$ increases, the consumption of computing decreases gradually, but so does the accuracy. On balance, $r$=8 is a cost-effective choice.

## 5 CONCLUSION

In this paper, we propose a novel paradigm for video action recognition called "align before adapt" based on the Visual-Language-Pretrained (VLP) model. Our approach achieves fine-grained visual-semantic alignment by incorporating a token merging strategy in VLP image encoders, enabling the perception of region-aware visual appearance. The embeddings of these regions are then matched with the text corpus of action-related entities using embedding similarity. The aligned entities are further leveraged in video representation adaptation, preserving the visual-semantic correspondences from VLP. Our paradigm demonstrates superior generalizability and competitive performance with low computational costs. Additionally, our framework is compatible with more powerful current and future visual-language foundation models.

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

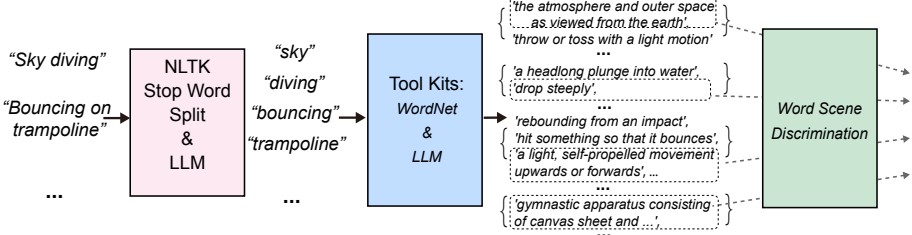

Figure A1: Snapshot of text corpus construction

## A    TEXT CORPUS CONSTRUCTION

In Fig. A1, we present a snapshot of the process of generating a text corpus from external video descriptions. For each description, (1) we extract the relevant action-related units via NLTK stop word removing and ChatGPT, where we design a prompt template *"What are identifying object/body parts/scenes/roles of an action {label}? List them concisely."* (2) We then use the WordNet tool to generate a sequence of explanatory descriptions for each extracted single-word unit. For the extracted phrase unit, we prompt ChatGPT to generate explanatory descriptions with the following templates: *"Concisely describe what an action {phrase unit} looks like"*, *"Concisely list potential explanations for {phrase unit}"*, and *"Concisely explain {phrase unit} in one sentence"*. (3) To determine the most appropriate description for each unit, we employ the Lesk algorithm and T5-based word sense disambiguation model according to the action labels. All of these procedures are automated through code that utilizes the sources mentioned in the manuscripts.

## B    TRAINING CONFIGURATION

In fully-supervised training, we set the batch size to 256 and adopt the AdamW optimizer with $\beta_1 = 0.9$ and $\beta_2 = 0.98$. The learning rate is $8\times10^{-6}$ for the ViT module in the region-aware image encoder and $8\times10^{-5}$ for the remaining learnable parts. The temperature $\tau$ of the softmax in Eq. 4 is set to 1. In few-shot experiments, the learning rate of the multi-modal video decoder is scaled up by ten times, and the batch size is reduced to 64. Regarding the text corpora, we initially constructed a text corpus based on Kinetics-400. When adapting to new scenarios, i.e. few-shot/zero-shot experiments, we reuse the collected entities and expand the text corpus with new action labels. All the text entities are embedded offline and fixed throughout experiments. For data augmentation, we utilize the technique including *RandomFlip, MultiScaleCrop, Mixup, and Label smoothing*, following the manner of X-CLIP (Ni et al., 2022).

## C    ADDITIONAL EXPERIMENTS

### C.1    FULLY-SUPERVISED EXPERIMENT ON KINETICS-600

Tab. C1 presents the results on Kinetics-600. Our ALT-B/16 outperforms MTV-L (Fan et al., 2021) by *0.5%* by using 32 frames per video with three views. Equipping with a larger backbone, ALT-L/14 achieves *88.5%* top-1 accuracy with computation consumption of only 4947 GFLOPs, which takes the lead among the methods that adopt similar-level pre-trained models and data.

Table C1: Comparison on Kinetics-600.

| Method | Pretrain | Top-1 | GFLOPs | Views |
|---|---|---|---|---|
| MViT-B-24 (Fan et al., 2021) | - | 83.8 | 1180 | 32×5×1 |
| VideoSwin-L(384↑) (Liu et al., 2022) | IN-21k | 85.9 | 25284 | 32×4×3 |
| ViViT-H/16x2 320 (Arnab et al., 2021) | JFT-300M | 83.0 | - | 32×4×3 |
| ViViT-H/16x2 (Arnab et al., 2021) | JFT-300M | 85.8 | 48916 | 32×4×3 |
| TokenLearner-L/10 (Ryoo et al., 2021) | JFT-300M | 86.3 | 48912 | 32×4×3 |
| Florence(384↑) (Yuan et al., 2021) | FLD-900M | 87.8 | - | 32×4×3 |
| CoVeR (Zhang et al., 2021) | JFT-3B | 87.9 | - | 96×1×3 |
| MTV-L (Yan et al., 2022) | JFT-3B | 85.4 | 18483 | 32×4×3 |
| MTV-H (Yan et al., 2022) | WTS-17B | 89.6 | 45537 | 32×4×3 |
| X-CLIP-L/14 (Ni et al., 2022) | CLIP-400M | 88.3 | 7896 | 8×4×3 |
| ALT-B/16 | CLIP-400M | 85.9 | 1308 | 32×1×3 |
| ALT-L/14 | CLIP-400M | 88.5 | 4947 | 32×1×3 |

Table C2: Comparison on Something Something V2.

| Method | Pretrain | Top-1 | GFLOPs | Views |
|---|---|---|---|---|
| MViT-B-24 (Fan et al., 2021) | K-600 | 69.7 | 708 | 32×1×3 |
| ViViT-L (Arnab et al., 2021) | IN-21K+K-400 | 65.4 | 11892 | 32×1×3 |
| MTV-B(384↑) (Yan et al., 2022) | IN-21K+K-400 | 68.5 | 11160 | 32×3×4 |
| EVL-B/16 (Lin et al., 2022) | CLIP-400M | 62.4 | 2047 | 32×1×3 |
| ST-Adapter-B/16 (Pan et al., 2022) | CLIP-400M | 69.5 | 1955 | 32×1×3 |
| ALT-B/16 | CLIP-400M | 66.8 | 1308 | 32×1×3 |

## C.2 FULLY-SUPERVISED EXPERIMENT ON SOMETHING-SOMETHING V2

The Something-Something V2 dataset collects more than 220000 video clips that belong to 174 action categories, covering basic human actions with everyday objects. Compared to Kinetics-400, it requires more temporal reasoning. We evaluate our approach on Something-Something V2 under full supervision. The accuracies are reported in Tab. C2. Among the CLIP-based works, our method outperforms EVL (Lin et al., 2022), but it is inferior to ST-adapter (Pan et al., 2022), which utilizes interleaved heavier 3D Convolution modules. We attribute the key to handling such kind of motion-heavy datasets to elaborately designed temporal communication mechanisms, which inspire future directions of our work.

Table C3: Effect of subcollections of the text corpus.

| Text corpus | Fully. | 2-shot | 0-shot |
|---|---|---|---|
| $\varnothing$ | 81.6 | 56.1 | 66.3 |
| $\mathcal{S}^{body}$ | 81.9 | 60.4 | 73.1 |
| $\mathcal{S}^{object}$ | 82.2 | 62.7 | 74.8 |
| $\mathcal{S}^{scene}$ | 82.2 | 61.9 | 74.1 |
| $\mathcal{S}^{motion}$ | 81.8 | 59.7 | 72.4 |
| $\mathcal{S}^{all}$ | **82.3** | **64.0** | **76.7** |

Table C4: Effect 1D-Conv and SA modules in the video decoder.

| SA | 1D-Conv | Fully. | 2-shot | 0-shot |
|---|---|---|---|---|
| - | - | 81.8 | 62.4 | 75.6 |
| ✓ | - | 82.2 | 63.2 | 76.3 |
| - | ✓ | 82.1 | 62.7 | 76.1 |
| ✓ | ✓ | 82.3 | 64.0 | 76.7 |

## C.3 INVESTIGATION OF TYPES TEXT CORPUS

To further validate the effect of text corpus, we set a baseline model by replacing aligned semantic embeddings in Eq. 6 with random learnable queries. The result is reported in the first row of Tab. C3 (2-shot and 0-shot experiments take 32 frames per video as input). Moreover, we evaluate the effectiveness of each sub-collection of text corpus by categorizing the text entities into four groups: object, body parts, scenes, and primitive motion. We find that each category is helpful, and the models with all text entities further outperformed the baseline, especially in the 2-shot (*+8.9%*) and 0-shot (*+10.3%*) experiments. The results reveal that our categorized text entities are complementary to each other, and semantic alignments promise more robust visual representations when facing a severe lack of data.

## C.4 VIDEO DECODER COMPONENT ANALYSIS

In addition to facilitating cross-modal information exchange, another important role of the video decoder is enabling spatiotemporal signal communication. We further investigate the effects of 1D-convolution (1D-Conv) and self-attention (SA) modules in the video decoder, and the ablation results are shown in Tab. C4. We find that both of them are beneficial to the performance.

