# OpenReview forum: "Align before Adapt: Efficient and Generalizable Video Action Recognition with Text Corpus"
_ICLR.cc/2024/Conference — ICLR 2024 Conference Withdrawn Submission_

### Official Review · Reviewer_X9sb · 2023-10-24

**Soundness:** 3 good
**Presentation:** 2 fair
**Contribution:** 3 good
**Rating:** 6
**Confidence:** 4

**Summary:**

This paper proposes a new concept called “align before adapt” in contrast to the previous “adapt then align” when a pre-trained image encoder is extended for video.
When using VLP, prior works usually construct a video embedding from video frames (this is the “adapt” from image to video), then match it with a text embedding (this is “align”).
In contrast, this work first “aligns” frame features with text, then aggregates them to a video-level feature.

- The visual encoder shown in Fig.3(a) is a ViT with ToMe to produce a frame feature X_t for frame t.
- Align: Then X_t is used as a query to encode the semantic embedding Q_t from a text corpus (Fig.3(b)).
- Adapt: These frame features X_1, …, X_T are 1D convolved and attended by Q_1, …, Q_T to produce a video-level embedding z (Fig.3(c)).

Experiments of fully supervised setting demonstrate that the proposed method works better than or comparable to other state-of-the-art models, with much fewer computation cost.
For zero-shot and few-shot, the proposed method outperform others.

**Strengths:**

The proposed concept, “align before adapt”, is reasonable and interesting. This kind of approach is not so popular because 3D patches are usually better for video representation such as ViViT and VideoMAE. Recent CLIP-based video models uses and combines features of CLIP applied to each frame, work well for zero-shot and few-shot settings, while the main approach is first construct a video-level representation.

The approach of this work transforms CLIP features of frames first, before temporal aggregation. It might inspire other potential following works for video representation by using pre-trained image encoders.

**Weaknesses:**

The alignment of the “align before adapt” concept in this work heavily relies on the text corpus (p.4, sec.3.1). It seems to be versatile and reusable, however, hand-crafted and the performance may be affected by the quality of the corpus and the vocabulary size K. There are not discussions on the corpus quality, and K is not shown.

The corpus may help the model by providing cues of “objects” and “scenes”, and the appendix shows show to construct the corpus by asking “What are identifying object/body parts/scenes/roles of an action {label}?” to an LLM model. However it may boost the notorious representation bias (background bias or scene bias). If it is true, it should be discussed as a limitation of the approach using a text corpus constructed in this way.
Also datasets with less bias (such as SSv2 or Diving48) should be used in the experiments.

There is a trend using learnable prompts when a frozen encoder, and there are some works for action recognition, such as ViFi CLIP. This approach shows a resemblance to the proposed method in terms of transforming frame-wise features of a pre-trained image encoder before aggregation. This prompt learning approach is simpler and does not require the preparation of a hand-crafted text corpus, which could be an advantage over the proposed method.
- (ViFi CLIP) Fine-tuned CLIP Models are Efficient Video Learners, CVPR2023


The proposed “region-aware” image encoder (Fig3a) uses ToMe, however the meaning of “region-aware” is not well explained and hence not clear. It is simply using ToMe, but
in the experiments ToMe and region-aware tokens are separately used in Table 5. It should be clarify more about the token merging and the term “region-aware”.




Minor comments:

- ALT, the name of the proposed method, is not explained anywhere.
- ToMe appears p2, but not explained (full-spelled) until p4.
- Fig3(c), the final \hat{Q}_M looks the text semantic embeddings because of the symbol and the connection from Q in the left of Fig3(c). But it is actually the visual embedding as \hat{X} is used as value. It might be better to change the notation and figure.

**Questions:**

How much does the quality of the text corpus affect the results, and how much is the representation bias in this approach ?

What is the similarity with prompt learning approaches to modify frame-wise features before temporal aggregation ?

---

### Official Review · Reviewer_FRFQ · 2023-10-31

**Soundness:** 3 good
**Presentation:** 3 good
**Contribution:** 3 good
**Rating:** 6
**Confidence:** 4

**Summary:**

An "Align before adapt" pipeline is proposed in this manuscript. The method first align the frame-level features with fine-grained action related prompt to extract fine-grained features. Besides, the token merging technique and region-aware image encoding is also involved to boost the fine-grained representation extraction. The proposed method is evaluated on scene-biased datasets Kinetics, UCF-101 and HMDB-51. Extensive experiments are conducted to show the effectiveness and efficiency of the proposed method.

**Strengths:**

1. The proposed method achieves state-of-the-art performance, and reach good balance between effectiveness and efficiency.
2. The idea of utilizing action-related text corpus to extract fine-grained visual video representation is novel. I believe this idea could inspire the community to fully utilize corss-modality pre-trained model for many other vision tasks.
3. This paper is well-written and origanized.

**Weaknesses:**

1. Only the scene-biased datasets are evaluated in this manuscript. It would be better to also evaluate the proposed method on the temporal-reasoning datset sthv1, sthv2.
2. In table5, the performance improvement under Fully-supervised setting is limitted.
3. Minors: In left-bottom of Figure-right, there are two "Ours-B/16" shown.

**Questions:**

1. The idea of coupling action-related text corpus with region-aware image encoding is good to extract fine-grained representations. Though action-related prompt like motion is included in the text corpus, I am not sure whether the motion dynamic is deeply extracted by the prompting. The proposed method is evaluated on the static-scene-biased dataset like kinectic, UCF-101. I suggest the author to also test the method on sthv1 or sthv2.
2. I believe the similar idea could also be applied on image-related tasks, and I would suggest the author to also test the proposed method on image tasks like image recognition.
3. It would be interesting to extend the align-before-adapt strategy to temporal domain to see whether there is any performance improvement.

---

### Official Review · Reviewer_FfRS · 2023-10-31

**Soundness:** 3 good
**Presentation:** 3 good
**Contribution:** 3 good
**Rating:** 5
**Confidence:** 4

**Summary:**

In this paper, the authors present a novel paradigm titled 'align before adapt' for video action recognition, leveraging VLP models. The method integrates a token merging approach within the VLP image encoder and orchestrates alignment between the text corpus and region-sensitive visual cues. This alignment is then harnessed to enhance video representation learning. Demonstrating its efficacy, the proposed approach registers commendable results across different configurations.

**Strengths:**

The proposed method achieves competitive performances across various settings.

This paper is well-written and the organization of the paper is good.

**Weaknesses:**

1. While the authors delve deeply into text-corpus construction in Section 3.1 and the Appendix, the absence of an ablation study to underline its efficacy is noticeable.

2. The term 'fine-grained' in relation to alignment is somewhat ambiguous. It would be beneficial if the authors could elucidate why they perceive their alignment to be 'fine-grained'. From my perspective, it appears more akin to a cross-attention mechanism between image and text features.

3. There seems to be a discrepancy in the experimental results for ALT-B/16 (8-frame) between Table 1 and Table 5.

4. Out of curiosity, I wonder how performance metrics would shift if the token merging approach was incorporated into other benchmarked methods.

**Questions:**

Please refer to 'weakness'

**Details Of Ethics Concerns:**

NIL

---

### Official Review · Reviewer_UeNy · 2023-10-31

**Soundness:** 3 good
**Presentation:** 2 fair
**Contribution:** 2 fair
**Rating:** 5
**Confidence:** 5

**Summary:**

The paper introduces a novel paradigm "align before adapt" for the adaptation of image-level models to a video-level. It is achieved by aligning the region-aware embeddings with the text corpus of action-related entities. As stated in the paper, this paradigm should preserve the visual-semantic alignment of visual-language pre-trained models during adaptation which allows better interpretability and generaliztion ability. This approach allows to achieve state-of-the-art results with less computational complexity. It also improves a zero-shot and few-shot compabilities of the model.

**Strengths:**

1) The proposed approach shows good performance in comparison to existing state-of-the-art approaches on large-scale action recognition (Kinetics). It also allows the reduction of the computational complexity.
2) Aligning with textual corpora enables better generalizability for zero-shot and few-shot settings. It is even more visible for a small number of examples, one and two. It is a very beneficial property for practical use cases in comparison to other models.

**Weaknesses:**

1) The paper has a fair overall presentation, however, some parts of the paper can be improved. Section 3.2 is not easy to follow and maybe some rewriting can help mitigate this problem. Despite that Figure 2 looks visually well, it lacks a detailed description of all the steps of the model. It is not helpful for the understanding of the approach. Most of the tables in the paper also don't have captions that discuss where to look and what to look in the tables. It is also not always obvious how methods are grouped in the tables. Is it the same split for all tables or a different one? So some work should be done to improve the presentation of the paper.
2) The proposed approach is highly based on the existing Token Merging approach and semantic alignment. The potential novelty comes from combining them into one approach. However, from Table 5 it seems that the most contribution to zero-shot and few-shot performance comes from just region-aware visual tokens while fully-supervised performance is already high enough without any components. It is very important to better study the contribution of components and have a better discussion. As of right now, it is not easy for the readers to clearly understand what is the contribution of each component exactly.
3) One of the main properties of the proposed approach, reducing computational complexity, comes from the trade-off of the number of tokens ("r" in Table 8) which is an internal parameter of Token Merging. Do other components contribute to this trade-off or it is solely the property of Token Merging? If it is just Token Merging then the first contribution of the paper is explained by Token Merging itself but not the overall approach.
4) There are also some concerns about the collection of textual corpora. It seems to be a little vague and not easy to reproduce if somebody does not or will not have access to ChatGPT. It is also not obvious why ChatGPT (or other chat-bot or language model) is used for such collection, what are the reasons behind this decision? Does it bring some bias that allows better generalization? From Table C3 it seems that textual corpora and its content are crucial for the best performance for zero-shot and few-shot settings. However, this table and discussion are not presented in the main paper which is not the best decision. I think it is very important to include them in the main paper.

**Questions:**

No other questions except those listed in the "Weaknesses" section.

---

### Official Review · Reviewer_njJR · 2023-11-02

**Soundness:** 2 fair
**Presentation:** 2 fair
**Contribution:** 2 fair
**Rating:** 3
**Confidence:** 5

**Summary:**

The paper proposes an approach to solve the problem of action recognition via align before adapt paradigm. The approach includes three modules: 1) Action Relation Text Corpus - Construction of knowledge base text corpus 2) Region-aware encodings - Use of token merging to increase throughput and 3) Representation Adaptation between frame and semantic embeddings.

**Strengths:**

-	The proposed approach shows good results on multiple datasets in few and zero-shot scenarios.

**Weaknesses:**

-	The novelty of the proposed approach is limited or not clear. Most of the modules are used directly from previous SOTA works as it is like ToME and GroupViT. Generation of text from ChatGPT can’t be considered as a novelty.  The Video Adaptation module hardly makes any improvement as evident from Table C4. The ablation table shows keywords Align, ToME and Region which are all under one section 3.2. In contributions (Introduction), it is mentioned token merging is a contribution whereas it’s directly used from paper ToME. The story is not clear.
-	(Section 1) Claim - Adapt then Align - Destruct visual-semantic correspondence of VLP models - This is a strong claim nowhere shown/verified qualitatively or quantitatively. For example, paper [1], has a better score on SSv2 (Table C2) which shows that it learns better temporal fine-grained features necessary to classify the video. Plus other models are not trained on fine-grained text corpus.
-	(Section 3.1) Generation of text corpus - It’s really confusing - Let’s consider two classes of UCF101 - Basketball and BasketBall Dunk. Looking at Figure 4, there would be almost similar text corpus generated for both classes. It will be focusing on fine-grained features but it’ll be categorizing both of the classes as similar with this logic. It’s not clear if this text corpus is actually helping or not.
-	(Section 3.2) Whole of this section is just explaining paper [1] and [2].
-	(Section 4) Table - 1 - The comparison is not fair. 3 temporal clips looks like dense sampling from video which means more information which is not a fair comparison  against paper with the same backbones - L/14. The improvement is negligible with a similar number of trainable parameters. Best scores of X-CLIP and X-CLIP B/16 not shown. X-CLIP B/16 outperforms the proposed approach.
-	(Section 4.4) Ablation study: (Table-5) It needs to be extended to show effectiveness of each single module. Table 6 - I’m not sure why it’s there. With whom it’s being compared against.

**Questions:**

Please see above.